# Auditory Neuropathy Spectrum Disorders: From Diagnosis to Treatment: Literature Review and Case Reports

**DOI:** 10.3390/jcm9041074

**Published:** 2020-04-10

**Authors:** Romolo Daniele De Siati, Flora Rosenzweig, Guillaume Gersdorff, Anaïs Gregoire, Philippe Rombaux, Naïma Deggouj

**Affiliations:** Department of Oto-Rhino-Laryngology and Head and Neck Surgery, Cliniques Universitaires Saint-Luc, Université Catholique de Louvain, 10 Avenue Hippocrate, 1200 Brussels, Belgium; flora.rosenzweig@student.uclouvain.be (F.R.); guillaume.gersdorff@student.uclouvain.be (G.G.); anais.gregoire@uclouvain.be (A.G.); philippe.rombaux@uclouvain.be (P.R.); naima.deggouj@uclouvain.be (N.D.)

**Keywords:** ANSD, auditory neuropathy spectrum disorder, auditory synaptopathy, hidden hearing loss, genetics, cochlear implant

## Abstract

Auditory neuropathy spectrum disorder (ANSD) refers to a range of hearing impairments characterized by deteriorated speech perception, despite relatively preserved pure-tone detection thresholds. Affected individuals usually present with abnormal auditory brainstem responses (ABRs), but normal otoacoustic emissions (OAEs). These electrophysiological characteristics have led to the hypothesis that ANSD may be caused by various dysfunctions at the cochlear inner hair cell (IHC) and spiral ganglion neuron (SGN) levels, while the activity of outer hair cells (OHCs) is preserved, resulting in discrepancies between pure-tone and speech comprehension thresholds. The exact prevalence of ANSD remains unknown; clinical findings show a large variability among subjects with hearing impairment ranging from mild to profound hearing loss. A wide range of prenatal and postnatal etiologies have been proposed. The study of genetics and of the implicated sites of lesion correlated with clinical findings have also led to a better understanding of the molecular mechanisms underlying the various forms of ANSD, and may guide clinicians in better screening, assessment and treatment of ANSD patients. Besides OAEs and ABRs, audiological assessment includes stapedial reflex measurements, supraliminal psychoacoustic tests, electrocochleography (ECochG), auditory steady-state responses (ASSRs) and cortical auditory evoked potentials (CAEPs). Hearing aids are indicated in the treatment of ANSD with mild to moderate hearing loss, whereas cochlear implantation is the first choice of treatment in case of profound hearing loss, especially in case of IHC presynaptic disorders, or in case of poor auditory outcomes with conventional hearing aids.

## 1. Introduction

Auditory neuropathy spectrum disorder (ANSD) refers to a range of hearing dysfunctions characterized by compromised signal processing along the auditory nerve or by deficient transmission of this signal to the auditory nerve by the presynaptic inner hair cells (IHCs). Its diagnosis is in part based on evidence of altered neural processing such abnormal auditory brainstem responses (ABRs), with a reduced or absent wave V, despite evidence of preserved outer hair cells (OHCs) responses, such as otoacoustic emissions (OAEs) and/or cochlear microphonic (CM). [1,2,3,4]. The affected subjects present important speech discrimination difficulties, particularly in background noise, that are out of proportion compared to their pure-tone detection thresholds [5]. Impairments in the coding of temporal features of acoustic stimuli seem to be the main underlying mechanism responsible for these difficulties. These temporal encoding deficiencies may also account for the frequently limited benefits of acoustic hearing aids [6].

ANSD was first described by A. Starr and colleagues in 1996 in ten patients presenting with evidence of normal OHCs function but impaired neural transmission in the peripheral auditory system [7]. The term of “auditory neuropathy” was then proposed to initially describe hereditary sensorimotor neuropathies [8]. Later, several authors suggested the more comprehensive term of “auditory neuropathy/auditory dys-synchrony” (AN/AD) or “auditory neuropathies” to underline the loss of temporal coding due to an abnormal synchronization at different levels of the peripheral neural pathways [7,9,10,11].

More recently, studies of genetics and molecular biology on animal models have shown a wide range of localization of the sites of impaired functioning, ranging from the area of inner hair cells (IHCs) synapses to the auditory neural fibers [3,12,13]. As a consequence, the term “auditory synaptopathy” has been used to distinguish ANSD due to a dysfunction of the IHCs ribbon synapses from the “auditory neuropathy” due to neural fibers dysfunction [13,14,15,16,17].

The clinical profiles of ANSD are largely heterogeneous, according to the variety of possible etiologies. ANSD may results from both syndromic and non-syndromic genetic anomalies, environmental causes, as well as structural anomalies. Among these, cochlear nerve hypoplasia or aplasia, variable lesion sites and congenital-neonatal or lately acquired dysfunctions have been described.

Special attention must also be paid to the concept of hidden deafness or hidden hearing loss (HHL), which corresponds to a hearing impairment in which speech discrimination in challenging auditory conditions (noisy and/or reverberant environment, less or rapidly articulated speech) is abnormally impaired as patients also display normal pure-tone and speech audiometry thresholds in quiet. HHL is associated with well-synchronized ABRs. This auditory processing disorder may be caused by a specific synaptopathy-neuropathy, limited to auditory fibers responding to high intensity sounds with preservation of fibers responding to low intensity sounds [18,19]. This special kind of auditory synaptopathy-neuropathy will be discussed separately because it does not correspond to the currently accepted definition of ANSD where ABRs waves and speech audiometry in quiet are severely deteriorated [20].

This article provides a literature review on ANSD management with a special focus on its etiologies, audiological assessment and rehabilitation. In order to describe the variability of physiopathological and clinical features of ANSD, we used a conceptual approach based on the site of the lesion, from the more distal to the proximal site of the sensorineural pathway. Moreover, few own case reports have been described.

## 2. Prevalence of ANSD

The prevalence of ANSD remains uncertain, and studies report prevalences ranging from less than 1% of hearing impaired patients up to 10% [1,12,14,21,22,23]. This variability reflects the wide heterogeneity of clinical profiles of ANSD patients across studies [12].

Newborns discharged from a neonatal intensive care unit (NICU), about 10%–15% of newborns, have a higher prevalence of sensorineural hearing loss (SNHL), and particularly of ANSD [24,25]. The prevalence of SNHL in the NICU discharged population is around 1/50 compared to 1/1000 in normal-term newborn children [26]. Similarly, their ANSD prevalence is also higher compared to normal-term infants, as ANSD accounts for up to 30% of all SNHL in NICU discharged children [27,28,29,30].

## 3. Etiologies

A run-through of the normal process of cochlear transduction of an auditory signal into an electrical impulse transmitted along the auditory nerve is essential for understanding the underlying mechanisms implicated in ANSD as well as the heterogeneity of its clinical manifestations.

The IHCs are mechanosensory cells that convert mechanical deflections of the hair bundle at their apex into a molecular signal suitable for triggering the electrical firing of neuronal fibers. The deflection of the hair bundle of IHCs after the vibration of the tectorial membrane generates a cation influx in the IHCs body. The consequent depolarizing potential allows a calcium influx through a single or two voltage-gated calcium channels. The coupling of Ca^2+^ channels at the presynaptic site of the ribbon synapse triggers a high rate synaptic vesicles fusion, leading to the release of glutamate through the synaptic cleft. The presynaptic signal is transferred in a graded manner in order to respect the rate code of the spike of the fibers of the spiral ganglion neurons (SGNs).

High levels of neurotransmitter liberation in the synapse enable the generation of an excitatory postsynaptic potential that leads to the activation of a particular type of Ca^2+^ sensitive receptors, the AMPA receptors, leading in turn to the generation of the neural spike at the level of SGNs fibers, that travel along the axon to the SGN cell body. The effective encoding of auditory stimuli relies on rapid and precise spike initiation in auditory nerve fibers. Preservation of the graded presynaptic signal, of the high-rate kinetics of synaptic release and of the rapid restoration of postsynaptic membrane activation, are primordial for the precise encoding of temporal features of auditory stimuli. Effective frequency phase-locking of firing in the SGNs fibers to the frequency of the auditory signal is also an essential element for pitch encoding of pure-tone sounds up to 1 kHz in frequency [31,32]. Furthermore, high densities of voltage-gated ion channels, such as sodium (Na_v_1.1, Na_v_1.6), potassium (K_v_1.1, K_v_2.2, K_v_3.1b, K_v_7.2 and K_v_7.3), ankyrin-G and Caspr channels on the auditory fibers, also contribute to the accurate coupling of neural firing to the synaptic input [33]. From the synapse with IHCs, peripheral axons of SGNs proceed in the modiolus of the cochlea and continue as proximal axons towards the midbrain. SGNs are bipolar neurons, and frequency-tuned to a specific inner hair cell in order to maintain the tonotopy during auditory stimuli processing [34]. The precise temporal coding is possible due to a graded-fashion ionic influx along the axons, in particular of Na^+^ [35].

A dysfunction at any level of this complex transduction machinery may disturb the coding of acoustic features, particularly of temporal cues. The potential sites of lesion and dysfunctions are various: IHCs, synapses between IHCs and type I auditory fibers, neural fibers, synapses between the fibers and their targets in the cochlear nucleus [36,37,38,39,40,41,42,43].

A wide range of etiologies has been reported, including prenatal factors (genetics, which are discussed below; cochlear nerve malformation; fetal infection like measles, mumps or cytomegalovirus—CMV; dysmaturity); postnatal factors (genetics with delayed onset of symptoms, prematurity, perinatal disorders such as severe icterus and kernicterus, hypoxia with mechanical ventilation, septicemia, ototoxic drugs, meningitis) [12,42,43].

In Figure 1, we report the audiological assessment of a 5-year-old child with ANSD caused by neonatal hypoxia. Hearing aids were fitted at the age of 8 months. At first, his babbling developed correctly. At the age of 2, he emitted low- and high-pitched phonemes, whereas an improvement of language skills in the lexical and morphosyntactic fields was observed at the age of 3. At that time, discrimination of environmental noise (such as animal sounds) remained poor. Besides speech therapy support, teaching aids have been implemented. Hearing performances are still fluctuating and remain poor in noise environment.

Moreover, studies reporting temporal bone analysis of premature and full-term infants admitted to the NICU showed specific loss of IHCs in respectively 27% and 3% of the cases [27]. Magnetic resonance imaging of auditory pathways have also shown structural abnormalities in up to half of healthy newborn with ANSD [22]. Children fed on a poor thiamine diet leading to a thiamine-deficiency may present with ANSD, which can be successfully treated with supplementary thiamine [44]. In this case, individuals may show the peculiar pattern of preserved OAEs and abnormal ABRs, but absent OAEs have also been reported in some cases, suggestive of a simultaneous impairment of OHCs.

Most deficits caused by lesions of other cochlear components, such as tectorial membrane, OHCs, IHCs, supporting cells or stria vascularis impairments, have not been associated with ANSD-like expression because they impact not only IHCs but also OHCs functioning and respect a certain degree of neural synchronization.

### 3.1. Genetic Etiologies

Various potential genetic causes of ANSD have been reported [12,13]. A genetic mutation may concern the synapses, the neural fibers or both sites. The distinction between these various lesion sites is essential when studying genetic causes of ANSD, since their impact on hearing outcomes differs, and may also explain variability in cochlear implantation (CI) outcomes. By directly stimulating the cochlear neural fibers, CI allows the bypassing of disorders involving IHCs synapses [13]. In case of a more central postsynaptic lesion involving the SGNs or more ascendant axons, the bypass implemented by the CI may be insufficient, yielding lower hearing performances.

#### 3.1.1. Presynaptic Synaptopathies

The first identified genetic cause of ANSD has been a mutation of the *OTOF* gene encoding for Otoferlin, a protein of the Ferlin family involved in the mechanism of presynaptic membrane fusion [45]. Otoferlin plays the important role of calcium sensor at the IHCs presynaptic site, as it binds calcium and phospholipids at the final step of exocytosis of glutamatergic vesicles at the ribbon synapse [37,46,47,48,49,50,51].

Mutations of *OTOF* are responsible for a form of nonsyndromic autosomal recessive sensorineural deafness, DFNB9, defined by a severe to profound congenital or prelingual hearing impairment but preserve vestibular functions [52,53,54]. Patients typically display normal OAEs but abnormal ABRs [45]. However, *OTOF* mutations may produce heterogeneous clinical profiles with variable speech discrimination scores and electrophysiological findings.

Some mutations in *OTOF* are responsible for a type of hearing loss sensitive to temperature [53,55]. When febrile, patients show a severe to profound hearing loss, while at normal body temperature they present normal hearing or a mild hearing impairment affecting speech recognition, particularly in background noise [56,57,58]. Most the *OTOF* subjects demonstrate good hearing outcomes with CI [59,60].

According to the Deafness Variation Database, by the end of the year 2019, 7894 identified variants of *OTOF* mutations have been reported, 113 of which accounted for being pathogenic, 17 were likely to be pathogenic and 6298 remained of unknown pathogeny [61]. *OTOF* mutations were found in 2.4% of individuals with hearing loss [62]. Virally mediated gene therapy has been put forward as a promising prospect in restoring hearing in knock out mice with impaired *OTOF* [63].

Other gene mutations involving presynaptic proteins may potentially cause ANSD.

*CACNA1D* gene also plays an important role at the IHCs presynaptic sites [64]. It codes for a subunit of Ca_v_1.3, a Ca^2+^ channel that works as the trigger for the glutamate release at the synaptic site [65]. Since these channels are widely distributed across different compartments such as in OHCs, IHCs, cardiomyocytes, neuroendocrine cells and neurons, this mutation may produce a syndrome called “sinoatrial node dysfunction and deafness” (SANDD syndrome) in mice and in humans [64,66]. Severe to profound hearing loss is a common finding in these patients. These ion channels seem to also play a role in calcium-mediated oxidative stress leading to age-related hearing loss in male mice [67] and in inner ear differentiation [68].

*CABP2* is another presynaptic protein that interacts with voltage-gated Ca_v_ channels for the regulation of Ca^2+^ influx at the presynaptic site [69,70], leading to the vesicular release of synaptic glutamate. Schrauwen et al. reported an autosomal-recessive nonsyndromic deafness in a family presenting a mutation in the *CABP2* gene at the *DFNB93* locus, with a flat or cookie-bite moderate to severe prelingual hearing impairment and with Marfanoid features [71].

*SLC17A8* gene codes for the vesicular glutamate transporter type 3 (VGluT3), a protein responsible for the glutamate uptake at the IHCs synapse [72,73].

In humans, an early report of a 12q22-q24 deletion associated with congenital deafness [74] was later identified as an autosomal dominant nonsyndromic deafness at the *DFNA25* locus associated with mutations of *SLC17A8* [75,76]. Affected individuals show a progressive sensorineural hearing loss at high frequencies.

In animal models, mutations in *SCL17A8* have been held responsible for a disruption of the synaptic glutamate exocytosis form of ANSD, leading to a lack of excitatory synaptic transmission to terminal dendrites synapses. Seizures may be associated with hearing loss, as an expression of central nervous system damage [72].

However, good hearing outcomes have been observed after cochlear implantation in individuals with *SLC17A8* mutations, reinforcing the belief of a dysfunction at the synaptic site [77].

Virally mediated gene therapy has been shown to be a promising prospect in restoring hearing in knock out mice with impaired VgluT3 function [78].

#### 3.1.2. Postsynaptic Synaptopathies

The optic atrophy 1 gene (*OPA1*) codes for a mitochondrial protein [79,80] that plays an important role in mitochondrial stability and energy output shaping [81,82].

Mutations in *OPA1* result in progressive loss of visual acuity or in legal blindness. Visual impairment may occur as an isolated clinical finding in the nonsyndromic dominant optic atrophy (*DOA*) form or in the syndromic dominant optic atrophy (*DOA+*) form associated with hearing impairment [83,84,85,86].

*DOA+* occurs as a simultaneous onset of progressive visual impairment due to optic atrophy and of an auditory postsynaptic hearing impairment, due to the degeneration of the terminal axons of SGNs. Accordingly, OAEs and the receptor responses are preserved, but ABRs are abnormal [86].

Pure-tone audiometry shows moderate to severe hearing loss with high variability of the involved frequencies among individuals within the same family. Individuals with *DOA+* display in 60% of cases a syndromic association of hearing impairment, sensorimotor neuropathy, myopathy and ataxia [85,86,87].

ANSD may potentially be observed as an effect of other gene mutations involving postsynaptic proteins.

*ROR1* gene codes for a protein localized at the plasma membrane, the receptor tyrosine kinase-like orphan receptor 1. *ROR1* plays an important role in the NF-κB pathway for neural outgrowth. Mutated variants of the gene in animal models have been correlated with deficiency of SGN axons and a lack of innervation of the sensory hair cell synapses.

This mutation has been identified in two children from a consanguineous Turkish family presenting profound SNHL, whose OAEs were preserved. Therefore, these combined animal and human findings are suggestive of an auditory postsynaptic synaptopathy [88].

The *ATP1A3* gene codes for the α3-subunit of the transmembrane Na/K-ATPase pump, implicated in the regulation of intra- and extra-cellular ion levels [89].

Individuals with mutated variants of the *ATP1A3* gene present a syndromic phenotype including cerebellar ataxia, areflexia, pes cavus, optic atrophy and SNHL, summarized in the acronym CAPOS syndrome [90]. In the ten families where CAPOS syndrome has been described to date, only the specific heterozygous mutation c.2452G > A in exon 4 has been consistently reported [91,92,93,94,95,96,97,98].

Typically, affected individuals present slowly progressive ANSD and optic atrophy, but recurrent acute exacerbation of the impairment from infancy has also been described. Moreover, hearing and visual impairments may be associated with extensive acute neurological deterioration accompanied by ataxia, areflexia, hypotonia, lethargy and ophthalmoplegia, with a possible partial post-crisis recovery. It has been hypothesized that stressful events, such a febrile illness, may trigger the onset of such acute episodes [95].

Pure-tone audiograms display various degrees of sensorineural hearing loss from moderate to severe, along with poor speech discrimination. Distortion product Otoacoustic Emissions (DPOAEs) responses are preserved, whereas ABRs are absent or abnormal. In accordance with the postsynaptic role of *ATP1A3* encoded protein, CAPOS syndrome is listed among the auditory synaptopathies with a postsynaptic site of the lesion [99]. Overall good CI outcomes were reported [95] (Figure 2. For full medical history of this case, see reference 95).

*DIAPH3* gene codes for the diaphanous homolog 3 (*DIAPH3*), whose function at the synaptic and neural sites remains unclear [100,101]. Clinical and electrophysiological findings along with the good results obtained after cochlear implantation suggest a nonsyndromic autosomal dominant auditory neuropathy 1 (AUNA1) via a synaptic lesion, listing *DIAPH3* mutations as a postsynaptic neuropathy [102,103].

#### 3.1.3. Auditory Neuropathy

Auditory neuropathy is frequently associated with lesion of other peripheral neurons, leading to syndromic phenotypes [13].

Charcot–Marie–Tooth disease (CMT) is one of the most prevalent inherited sensori-motor neuropathies (HSMN), affecting approximately one in 2500 people in the United States [104]. Clinical findings include a progressive motor and sensory neuropathy, with a variability in inheritance, severity and neural damage localization, along with a SNHL with disproportionately poor speech perception compared to the loss expected from the cochlear impairment [105]. For this reason, CMT was included in the early description of auditory neuropathy [8]. Among several genes potentially involved in CMT, mutations in the *MPZ* gene and the *PMP22* gene have been correlated to the ANSD phenotype. Histological temporal bone analysis of CMT patients demonstrated pronounced SGNs fiber demyelination, whereas hair cells morphology was normal [106,107,108].

Speech perception scores of 54% after cochlear implantation in one subject with CMT suggest poor results in this case of auditory neuropathy [83].

Friedreich ataxia is an HSMN which is thought to be caused by similar damage at the SGNs level, resulting in auditory neuropathy [109].

The deafness–dystonia peptide-1/translocase of mitochondrial inner membrane 8A (DDP1/TIMM8A) is a protein involved in the transfer of metabolites into the mitochondrial inner membrane from the cytoplasm. A mutation of *TIMM8A* gene is responsible for an X-linked recessive progressive neurodegenerative syndrome associated with auditory neuropathy, named deafness–dystonia–optic neuropathy (DDON or Mohr–Tranebjaerg syndrome) [110].

Affected individuals also present dystonia and ataxia occurring in adolescence, progressive optic atrophy starting in the third decade, and dementia after the age of 40, reflecting their progressive degeneration of neurons [111,112].

*AIFM1* codes for a flavin adenine of the mitochondrial intermembrane space, the apoptosis-inducing factor mitochondria-associated-1, expressed in inner and outer hair cells and in SGNs. This protein has a role in oxidative phosphorylation and in the apoptosis pathway [113,114].

Mutations of the *AIFM1* gene are responsible for an X-linked auditory neuropathy associated with a progressive neuromuscular degeneration and cognitive decline, known as Cowchock syndrome. Clinical findings include numbness, unsteadiness and areflexia. In some patients with variants of the *AIFM1* gene, a delayed onset of cochlear nerve hypoplasia has been reported [115].

The mitochondrial asparaginyl-tRNA synthetase (*NARS2*) mutation has also been associated with auditory neuropathy and Leigh syndrome, an early-onset progressive neurodegenerative disorders characterized by symmetric, bilateral lesions in the basal ganglia, thalamus and brainstem.

Variants of *NARS2* due to homozygous missense mutation cause mitochondrial respiratory chain deficiency, leading to auditory neuropathy due to a cellular damage notably of the SGNs (DFNB94) [116].

#### 3.1.4. Synaptopathy and Neuropathy

Pejvakin, encoded by the *DFNB59* gene, is expressed in hair cells and SGNs, where it acts as a sensor and activates the autophagy mechanism to initiate the pexophagy or the degradation by peroxisomes, in case of oxidative stress such as noise-induced damage [117,118,119].

The initial description of *DFNB59* mutation-related deafness reported an ANSD phenotype with pathological ABRs and preserved OAEs. However, following reports showed a lack of both OAEs and ABRs in human and knock-in mice, due to a primary or secondary damage of the sensory compartment and to a higher vulnerability to noise-induced damage [118,120,121,122,123,124]. Interestingly, the viral transduction in mice with a deficiency of pejvakin has been demonstrated to completely restore the disrupted peroxisomes proliferation triggered by pejvakin mutations, preventing the damage from oxidative stress [117].

Other proteins expressed in the auditory pathway may potentially lead to ANSD.

The transmembrane serine protease 3 (*TMPRSS3*) is a protein broadly expressed across the human peripheral hearing pathways [125,126], notably in type II SGNs, and is involved in hair cells and spiral ganglion cells survival in animal models [126,127,128]. Mutations in *TMPRSS3* gene account for autosomal recessive SNHL with both a postlingual onset (*DFNB8*) or with the congenital onset of a severe to profound hearing loss (*DFNB10*) with normal CM and smaller auditory nerve neurophonic response, suggestive of an auditory neuropathy [129,130]. However, variable speech perception outcomes are reported after cochlear implantation in individuals with *TMPRSS3* mutations. Depending on the degrees of multisite lesions in various subjects, auditory results vary from good to poor [42,131,132,133].

## 4. Psychoacoustic Tests

Psychophysical evaluation of ANSD consists of adapted behavioral pure-tone audiometry and speech discrimination in quiet and noise; moreover, supraliminal testing such as gap detection and sound localization, tone decay and frequency discrimination should complete the audiological assessment. Investigation should include the evaluation of language skills, global cognitive and motor development in children, as well as attentional load and psychological profiles in adults.

### 4.1. Tonal and Speech Audiometry Thresholds

Evaluation of hearing impairment in ANSD patients with pure tone audiometry may show impairments ranging from normal to profound hearing loss with no specific pattern. Indeed, hearing loss profiles appear to be variable: from flat to more marked thresholds on low or high frequencies (Figure 3a).

ANSD is bilateral in about 75% of cases and unilateral in 25% [134]. The hearing thresholds frequently fluctuate and could reach variations of over 40 dB. Hearing fatigue may be present for high intensities and/or long-duration tonal stimuli and may impede the threshold investigation. Fluctuations are more frequent in children than in adults. Some children may show a spontaneous improvement in hearing reactions, more frequently in the first year after diagnosis [134], therefore special attention should be paid to tonal threshold evaluation in infants with ANSD caused by hyperbilirubinemia and anoxia [135]. Otherwise, a slow hearing deterioration over time is frequently observed in children and adults. Subjects with low frequencies hearing loss frequently show a deterioration overtime in their high and middle frequencies [134,136]. The tonal hearing thresholds are variables within subjects, even when accounted for by the same genetic mutation. For instance, subjects presenting an *OTOF* mutation present a profound deafness in 75% of cases, whereas it is severe in 22% and moderate in 3% [60]. Hearing fluctuations over time explain why repetitive hearing assessment is imperative. Associated cognitive impairments may also complicate the hearing thresholds estimation. The behavioral testing must be adapted to the psychomotor age in children and to the attentional effort abilities in adults.

Speech perception tests show very poor speech discrimination abilities, even in subjects with preserved tonal thresholds, as shown in Figure 3b. Impaired speech perception skills are typically out of proportion compared to the tonal thresholds. Background noise further deteriorates residual speech discrimination [1,7,134,137,138].

### 4.2. Supraliminal Tests

The supraliminal psychoacoustic tests study the auditory processing of complex sounds presented at a comfortable level and not only around the hearing thresholds.

When short-duration stimuli (200 ms) are used, the psychoacoustic measures show that the disrupted neural activity associated with ANSD has minimal effects on intensity-related perception, such as discrimination of loudness changes and sound localization using interaural level differences or on pitch discrimination at high frequencies [4].

In contrast, timing-related perception is impaired in ANSD subjects. They present poorer pitch detection for low frequency, indexing a disturbed phase-locking, and a less efficient detection of shorter stimuli and gaps between stimuli sounds. Moreover, they need larger changes to detect frequency modulations and have an abnormal susceptibility to the masking effect that lasts longer in ANSD, particularly for backward masking. Individuals with ANSD are disturbed largely by noises arriving after the stimuli.

The abilities of sound localization using interaural time differences are also decreased [4].

The processing of short acoustic stimuli in ANSD shows an inverse pattern of what is observed in nonsynaptic cochlear disorders, where intensity perception is impaired but temporal processing is generally preserved.

When long-duration stimuli are used, ANSD subjects display abnormal loudness adaptation, depending on the site of the lesion. If their abilities to detect brief changes of frequency or intensity in continuous sounds are investigated, subjects with ribbon synapses disorders (e.g., in mutations of otoferlin) report the disappearing of the background tone whereas the changes in intensity or frequency are correctly detected, contrary to neuropathic subjects that do not report a loss of the steady tone [139].

The perception of loudness of sustained tones presented for 3 minutes is relatively preserved and stable in normal hearing subjects, but it is decreased in ANSD as a function of frequencies. Ribbon synapses’ ANSD shows an abnormal and rapid auditory fatigue, for both long-duration low and high frequencies presented at comfortable levels, with loudness loss of more than 90% for 8 kHz at 90 s and around 50% for 250 Hz. Individuals with neuropathic ANSD present a normal adaptation to low frequencies but abnormal to high frequencies, even though less severe and less rapid than in ribbon disorders. Abnormal adaptation may be associated with depletion of neurotransmission that may exacerbate the abnormal speech perception in ANSD [54].

## 5. Objective Assessment of ANSD

Typical auditory patterns in ANSD include the preservation of OAEs and CM and absent or altered neural waves of the ABRs [41] by loss of neural response synchronization or generation. Therefore, objective assessment is essential in the correct diagnosis of ANSD. In Figure 4, we report the audiological assessment in a 42-year-old woman with ANSD and bilateral atrophy of cochlear nerve. Patient presents with Melkersson–Rosenthal syndrome, Ehler Danlos syndrome and Hashimoto’s thyroiditis. She suffers from recurrent uveitis. Despite preserved hearing in quiet, patient complaints of poor speech understanding in challenging hearing conditions, especially with background noise.

Besides OAEs and ABRs, a complete physiological assessment should also include impedance and stapedial reflex evaluation, as well as other event-related potentials, such as electrocochleography (ECochG), auditory steady-state responses (ASSRs) and cortical auditory evoked potentials (CAEPs).

OAE, first described by D. Kemp in 1978 [140], is the sound generated by the OHCs of the cochlea, which can be recorded after a transient stimulation (transient-evoked, TEOAEs) or two simultaneous pure-tone stimuli (distortion product, DPOAEs) with a sensitive probe placed in the external ear canal [141]. This sound represents the cochlear amplifier energy that originates from the somatic motility and the stereocilia bundle of OHCs, but multiple localizations and a more complex origins have been speculated [142].

OAEs may decrease or disappear in 20 to 80% of the subjects with time, particularly after wearing hearing aids [6,13,42]. Moreover, cases of recovering of ABRs synchronization in newborns with ANSD have been described [143]. This recovery at the peripheral level may not concern more central levels, resulting in supraliminal auditory processing disorders.

OHCs are the main contributors to the generation of an early evoked response, the cochlear microphonic (CM), a short-latency potential occurring before wave I of the ABRs. It has been shown that CM originates from the mechano-sensitive transduction channels in the stereocilia of both IHCs and OHCs, with a predominance for the OHCs due to their greater number, and is produced by the opening and closing of transduction channels in the hair bundle that follow the movement of the basilar membrane [144,145]. The CM is an electrophysiological response that is polarity dependent. CM is highlighted by the subtraction of the ABRs evoked by rarefaction and condensation stimuli. The summation brings out the summating potential (SP) and neural responses like the compound action potential (CAP) on ECochG or wave I on ABRs. Even when non-detectable at early neonatal stage, CM could appear later in preterm infants. In Figure 5, we report a case of a 26-week preterm newborn with ANSD of unknown etiology with no clinical history.

Transtympanic ECochG recorded at the promontory of the cochlea is a gold standard for the study of the CM, SP and CAP, due to the proximity of the electrode with the basal portion of the cochlea and a better signal-to-noise ratio compared to an extratympanic approach in which a probe is placed in the external auditory canal [146,147,148].

ECochG may also be recorded intraoperatively during cochlear implantation, by means of the intracochlear electrode of the implant, after acoustical stimulations [149,150,151].

Both OAEs and CM represent the physiological measures of the activity of OHCs. Therefore, the presence of CM is crucial for the diagnosis of ANSD.

The origins of SP are more debated and remain unclear, but it is accepted that SP mainly reflects the activity of IHCs [152]. More recently, the effects of ototoxins and neurotoxins in animal models suggested a possible combined receptor and neural origin of the SP [144].

Another component of ECochG responses reflecting the neural activity is the auditory nerve neurophonic (ANN), an auditory evoked potential representing the phase-locking activity of the neural unit of the auditory nerve to low-frequency stimuli [153,154]. It is difficult to distinguish from the cochlear microphonic.

In normal-hearing subjects, ECochG should consistently provide all the above-mentioned components.

The comparison of amplitudes of CM in normal hearing subjects and ANSD patients has shown no significant difference [155], although the duration of CM can be longer in subjects with ANSD [146].

In auditory synaptopathy, the CM and the SP are preserved, while CAP is not detectable [11,12] (Figure 1c). Most of the patients with ANSD display a prolonged negative deflection of ECochG responses without separation between SP and CAP potentials.

Patients with ANSD occurring with preservation of synaptic functioning may show normal SP and CAP components (Figure 6c,d). In Figure 6, we report the audiological assessment in a 48-year-old woman with acquired ANSD caused by bilateral cochlear nerve hypotrophia of unknown etiology, with no other medical history. She presents with a deterioration of speech discrimination despite partial preservation of tonal perception. Detection of environmental noises remains fair, but discrimination of sounds is absent.

However, the presence of neural activity recorded by ECochG is not sufficient for a diagnosis of ANSD [38,40,156]. In fact, even in presynaptic lesions, some residual apical connections in the cochlea may still be functional [43].

Therefore, the study of adaptation of the cochlear potentials is essential for the differential diagnosis of ANSD [157]. Adaptation means a decrease in the amplitude of the electrophysiological responses in the presence of stimuli repetition, representing a fatigue-like response.

Among the other evoked potentials, auditory steady-state responses (ASSRs) best predict the pure tone threshold in individuals with both normal and impaired hearing. ASSRs are evoked by rapid and periodical auditory stimulation using frequency-specific stimuli, varying from 0.5 to 4 kHz [158]. They may be present in ANSD subjects and produced by the phase-locking to microphonic and/or neurophonic responses [159].

Late auditory potentials recorded in children with ANSD have been correlated to cortical maturation and behavioral outcomes and may be used as a good indicator of the disruption of cortical development due to neural dys-synchrony [160].

Even with absent ABRs, cortical auditory evoked potentials (CAEPs) can be evoked in patients with ANSD [161,162]. Abnormal or absent CAEPs have been observed in approximately one-third of children with ANSD [139,160].

Among other CAEPs components, the P1 or N100 responses are significantly correlated with auditory skills development [163].

N100 responses provides a fairly objective estimation of the psychoacoustic threshold for gap detection and perceptual speech skills and may be useful for predicting significant benefits from hearing amplification in patients with ANSD [1,138,161,162,164].

N100 differences in amplitude and latency may suggest the presynaptic or postsynaptic site of the lesion. In presynaptic ANSD, N100 amplitudes in response to changes in frequency of 250 and 4000 Hz are larger than in postsynaptic disorders. Conversely, in postsynaptic ANSD, N100 latencies evoked by frequency changes show more delayed latencies, whereas normal latencies are evoked in individuals with presynaptic ANSD [139].

Mismatch Negativity (MMN) [165] and P300 (P3a and P3b) [166] are preattentive and attention-targeted auditory cortical responses, respectively, to rare and unpredictable (deviant) stimuli in repetitive sequences of standard sounds, called oddball paradigms. An example of those cognitive evoked potentials is shown in Figure 7e, as a part of the audiological assessment in a 25-year-old girl with ANSD and progressive Ponto-bulbar palsy syndrome (also called Brown–Vialetto–Van Laere syndrome). This rare degenerative familial disorder is caused by a riboflavin transporter deficiency. The patient presents with neural impairment of the 8th and 12th cranial nerves and the optic nerves. Without lip-reading, communication with others is possible only at low speech rate. Speech perception increases when lip reading is available. Hearing aids did not improve significantly her auditory perception skills. However, speech perception was good after two months of CI experience. 

Cranial magnetic resonance imaging and genetic evaluation are recommended to detect potential morphologic abnormalities of the cochlear nerve or peripheral and/or central nervous system (Figure 6f) [42].

## 6. Therapy and Outcomes

The current management of individuals presenting with ANSD varies according to the severity of the impairment. However, management remains challenging and is frequently tailored case-by-case. It is based on bottom-up (auditory skills restoration by hearing aids) and top-down procedures (hearing and speech training).

Generally, a multidisciplinary approach is favored.

Once the full workup is completed, two main therapeutic options may be offered. The first relies on maximizing signal to noise ratio to improve listening in noise. The second consists of sound amplification through conventional hearing aids or cochlear implant (CI).

Treatment modality should be chosen following intensive counselling of the patient or the pediatric patient’s family. Clear information must be provided regarding the eventual outcome limits in the case of postsynaptic or associated CNS lesions.

Signal-to-noise ratio maximization

In milder forms of ANSD, patients may benefit from the use of “FM-listening” devices. Indeed, speech comprehension in noise is one of the main difficulties encountered by ANSD patients, even in milder forms with little audiometric threshold impact. The use of “FM-listening systems“ has been extensively described in the classroom environment. Speech is detected directly from the speaker and transmitted through radio signal to headphone receivers worn by the child, enhancing the speech signal. Positive outcomes of this device have been described in several studies [167], as well as when used in conjunction with CI [3], with reported improvement of speech in noise comprehension [168].

Amplification: Hearing aids and cochlear implant

Conventional hearing aids may be proposed, although most studies report poor outcomes in the case of ANSD [8,169]. Indeed, conventional hearing aids amplify the signal but fail to overcome the neural dys-synchrony responsible for impaired speech comprehension. In a large multicentric study, Berlin et al. reported the outcome of hearing aid use in 85 patients with confirmed ANSD as well as 49 patients with CI. Hearing aids were described as offering “good benefit” by 3.53% of patients, “some benefit” by 10.59% of patients, “little benefit” by 24.71% of patients and “no benefit” by 61.17% of patients. Conversely, CI recipients reported 85% of successful rehabilitation in speech recognition, with an additional 8% of patients in the cohort being too recently implanted to obtain a correct evaluation of their performance [3]. Similar positive results regarding CI in ANSD management have been reported in both prospective and retrospective studies. Nevertheless, in the case of a more moderate hearing loss, nerve malformation and/or syndromic forms of ANSD, acoustic hearing aids may be sufficient for some ANSD children. [8,13,42,170,171]. However, if their auditory behavior and language skills do not develop normally or within the expected outcomes, CI must be provided [172].

Indeed, CI appears to be an effective rehabilitation modality for ANSD patients. This may be explained by the fact that the implanted electrode delivers synchronized electrical impulses directly to the auditory nerve, bypassing the presynaptic IHCs and its synapse involved in the unsynchronized firing of the auditory nerve described in ANSD [42].

Despite this evidence, CI outcomes in speech perception remain variable. Many biographical factors (residual hearing, age at implantation, auditory privation duration, cognitive and socioeconomic factors, anatomic morphology) have been reported to influence CI outcomes, but account for less than 25% of their variability [43]. In case of ANSD, the site of lesion along the auditory pathway seems to also have a prognostic significance [173]. In two consecutive papers, Shearer et al. reported that optimal CI outcome in ANSD with genetic etiology is reached in individuals affected by presynaptic (IHCs) or synaptic dysfunction as opposed to patients with distal auditory nerve lesions. Shearer argues that CI outcomes in ANSD may be partly predicted by the genetic site of lesion and their effect on spiral ganglion function. In his studies, four ANSD gene mutations responsible for presynaptic or postsynaptic cochlear dysfunction were examined (*OTOF*, *SLC17A8*, *CACNA1D*, *CABP2*). Patients with these mutations had a significantly better outcome in speech recognition than patients with mutations in the four studied genes responsible for spiral ganglion dysfunction (OPA1, *DFNB59*, *AIFM1*, *DIAPH3*) [13,77].

The general rules for cochlear implantation must be promoted in children with isolated forms of congenital or very early onset ANSD: early implantation must remain the standard of care in case of severe to profound hearing loss with normal cochlear nerve anatomy. Otoferlin mutations are associated with excellent CI outcomes and typically on par with individuals with genetic mutations affecting the sensory partition of the cochlea [13]. Cochlear nerve malformation or associated disorders are negative prognostic factors for CI outcomes but not contra-indications, and speech perception impairment may be severe enough to indicate cochlear implantation [43] (Figure 8 and Figure 9).

## 7. Conclusions

The concept of ANSD is based on an operational definition: speech discrimination disorders and/or abnormal language development out of proportion with sound detection abilities, as well as preserved outer hair cell responses with very disturbed ABRs. ANSD may be caused by cochlear presynaptic and postsynaptic lesions or by lesions to ascendant cochlear fibers from the auditory nerve to the brainstem. Genetic, environmental, infectious, inflammatory and idiopathic causes may trigger the symptoms since birth, during childhood or later in life. The clinical patterns of ANSD are very heterogeneous. Hearing rehabilitation is more efficient in peri-synaptic disorders thanks to cochlear implants. However, genetic studies that have proven to be essential in the knowledge of underlying mechanisms of ANSD represent a promising therapeutic approach in the management of ANSD.

## Figures and Tables

**Figure 1 jcm-09-01074-f001:**
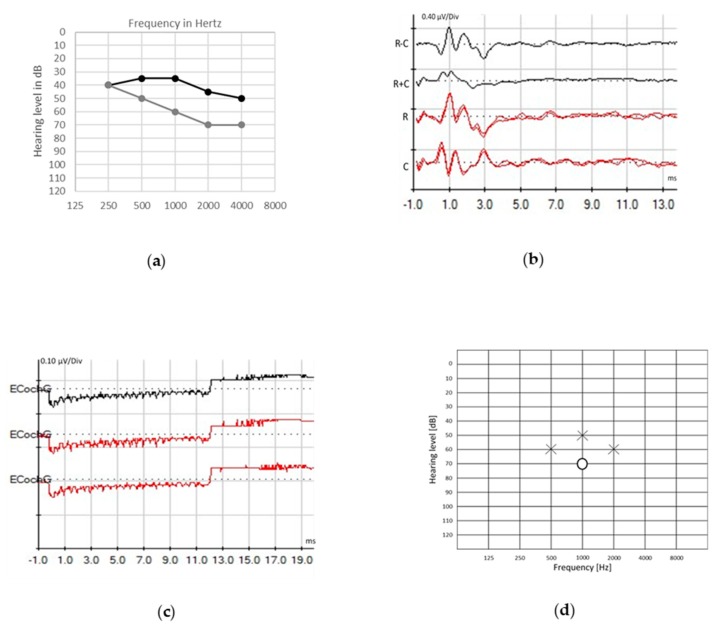
Audiological assessment in a 5-year-old child with auditory neuropathy spectrum disorder (ANSD) caused by neonatal hypoxia. Poor unaided pure-tone perception (**a**, gray line) was restored with hearing aids ((**a**), black line). Panel (**b**) displays auditory brainstem responses (ABRs) evoked by clicks presented in rarefaction (R) and condensation (C) polarities and the subtraction and summation of R and C (R − C and R + C, respectively), showing detectable cochlear microphonic (CM) and absence of waves V. Electrocochleography (ECochG) recorded through a transtympanic electrode on the promontory wall using 1000 Hz tone burst presented in alternated R and C polarities at a rate of 14.3 s (**c**) shows a large summating potential (SP). Auditory steady-state responses (ASSRs) thresholds (**d**) are present in the left (X) more than in the right (O) ear.

**Figure 2 jcm-09-01074-f002:**
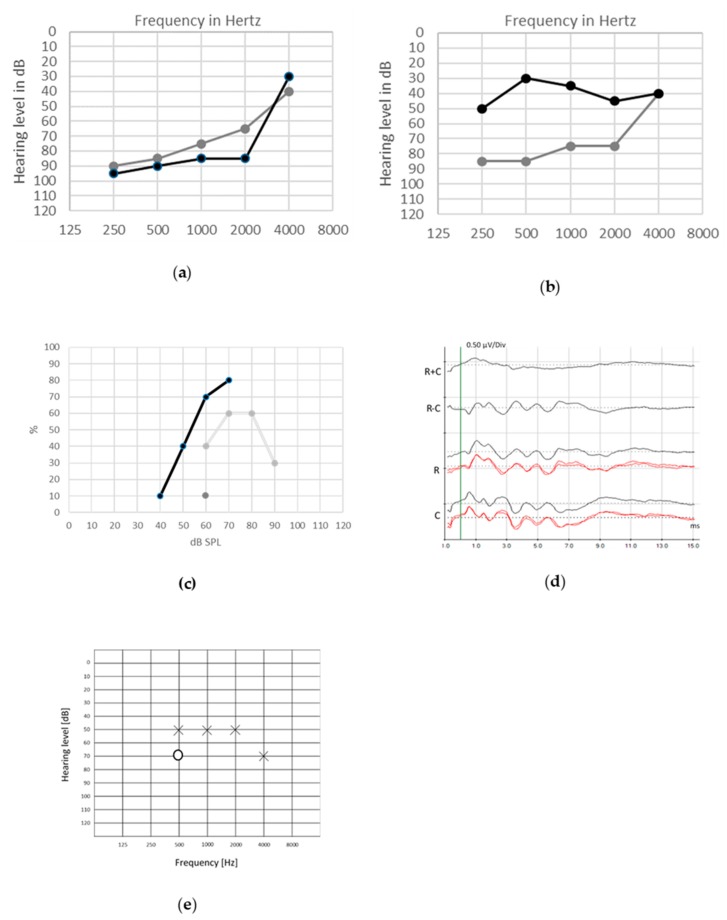
Audiological assessment in 8-year-old patient with CAPOS syndrome. CAPOS is an acronym for Cerebellar ataxia, Areflexia, Pes cavus, Optic atrophy and Sensorineural hearing loss. Unaided pure tonal thresholds in both ears are shown in (**a**). The tonal hearing thresholds remain poor in the right ear aided by an acoustical hearing aid ((**b**), gray line) but are clearly improved in the left ear by cochlear implant ((**b**), black line). Speech perception was poor in unaided condition ((**c**), 10% gray dot at 60dB), it was partially improved by acoustical hearing aids ((**c**), gray line) but became significantly better with cochlear implant ((**c**), black line). abnormal auditory brainstem responses (ABRs) (**d**) show the responses evoked by clicks presented in rarefaction (R) and condensation (C) phase and the subtraction of R and C (R − C), highlighting the cochlear microphonic (CM) and the lack of waves V in the summation (R + C). Auditory steady-state responses (ASSRs) thresholds (**e**) are present in the left (X) more than in the right (O) ear.

**Figure 3 jcm-09-01074-f003:**
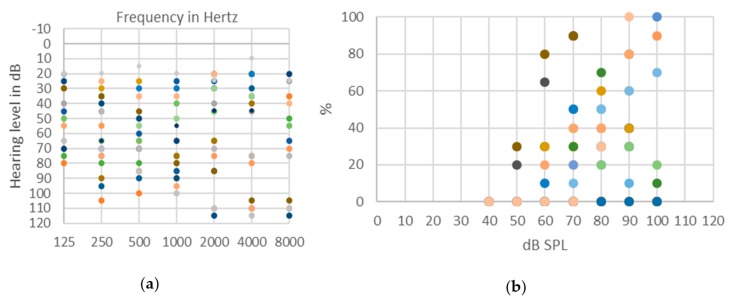
Distribution of pure-tone audiometry threshold (**a**) and (when possible) speech audiometry threshold for disyllabic words (**b**) in 14 patients aged between 5 and 48 years with auditory neuropathy spectrum disorder (ANSD). Etiologies of ANSD vary among patients. The figure is intended to show the large variability in hearing ability among patients with ANSD.

**Figure 4 jcm-09-01074-f004:**
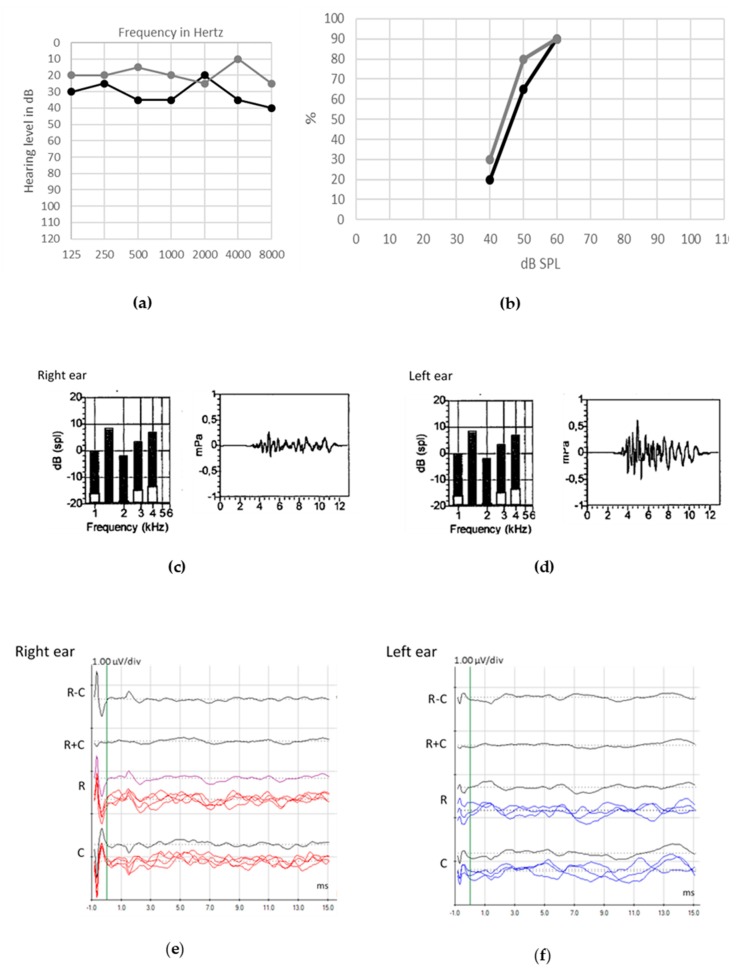
Audiological assessment in an adult patient with auditory neuropathy spectrum disorder (ANSD) and bilateral cochlear nerve atrophy. Unaided thresholds for pure-tone audiometry ((**a**), black line for right ear, gray line for left ear) and speech audiometry ((**b**), black line for right ear, gray line for left ear) in quiet are good. Otoacoustic emissions (OAEs) are present bilaterally in both temporal/intensity recordings (left panel in (**c**) for right ear, left panel in (**d**) for left ear) and spectral analysis (right panel in (**c**) for right ear, right panel in (**d**) for left ear); on the other hand, auditory brainstem responses (ABRs) are abnormal with no clear cochlear microphonic (CM) ((**e**) for right ear, (**f**) for left ear).

**Figure 5 jcm-09-01074-f005:**
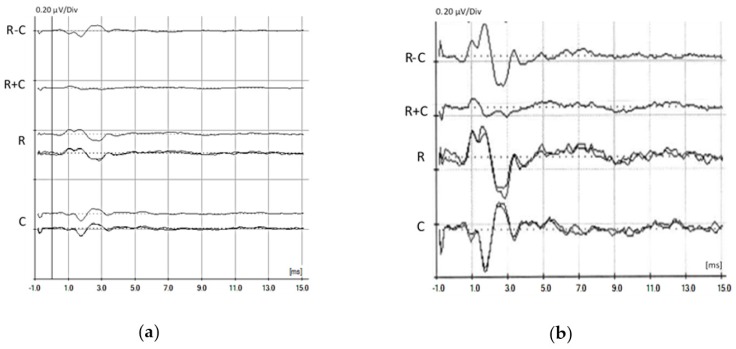
Auditory brainstem responses (ABRs) in a 26-week preterm newborn with auditory neuropathy spectrum disorder (ANSD). Three-month ABRs (**a**) show small early components, whereas a cochlear microphonic (CM) was detectable 9 months later (**b**), suggesting a late, although partial, maturation. Auditory steady-state responses (ASSRs) remain absent in both ears.

**Figure 6 jcm-09-01074-f006:**
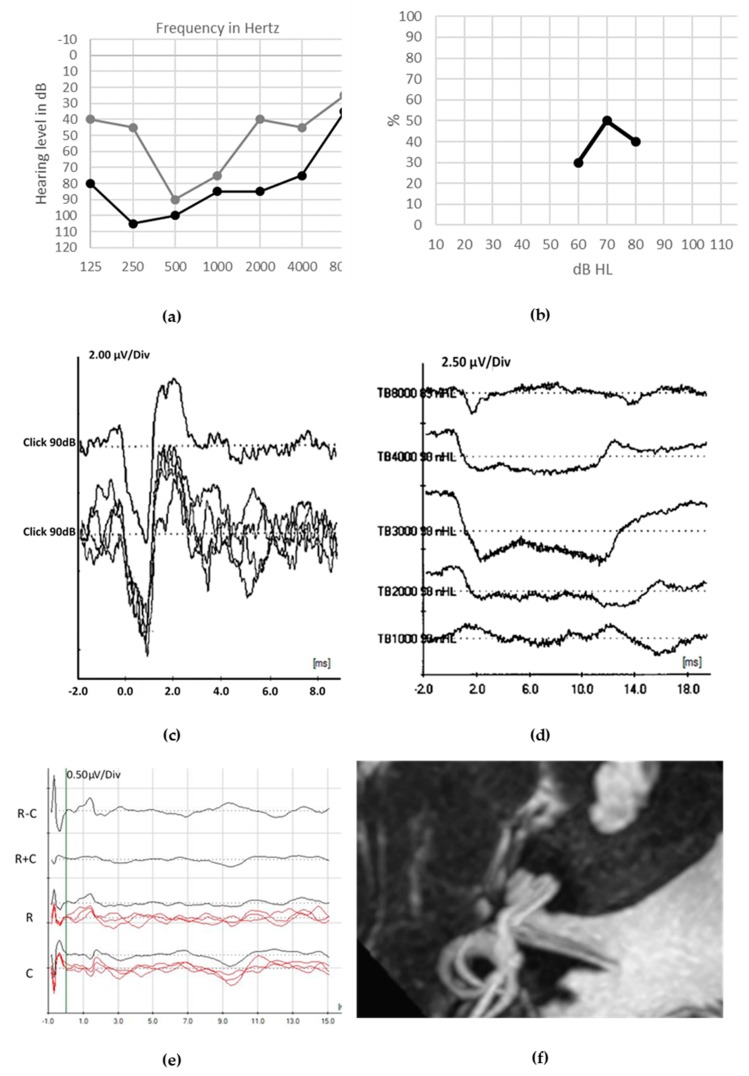
Audiological assessment in adult with acquired auditory neuropathy spectrum disorder (ANSD) caused by bilateral cochlear nerve hypotrophia of unknown etiology. Unaided pure tone ((**a**), black line for right ear, gray line for left ear) and speech ((**b**), both ears in free field) thresholds are poor, with no improvement with hearing aids. Electrocochleography (ECochG) recorded after clicks at 90 dB and at a rate of 14.3 s ((**c**), grand average, above; superimposed, below) shows preserved summating potential (SP) and compound action potential (CAP). ECochG responses to tone-burst stimuli at different frequencies ((**d**), grand averages for 8, 4, 3, 2 and 1 kHz) show a large SP. Auditory brainstem responses (ABRs) are absent, except for the cochlear microphonic (CM) (**e**). The 3-Tesla magnetic resonance imaging of the right ear without contrast (**f**) shows the hypoplasia of the cochlear nerve.

**Figure 7 jcm-09-01074-f007:**
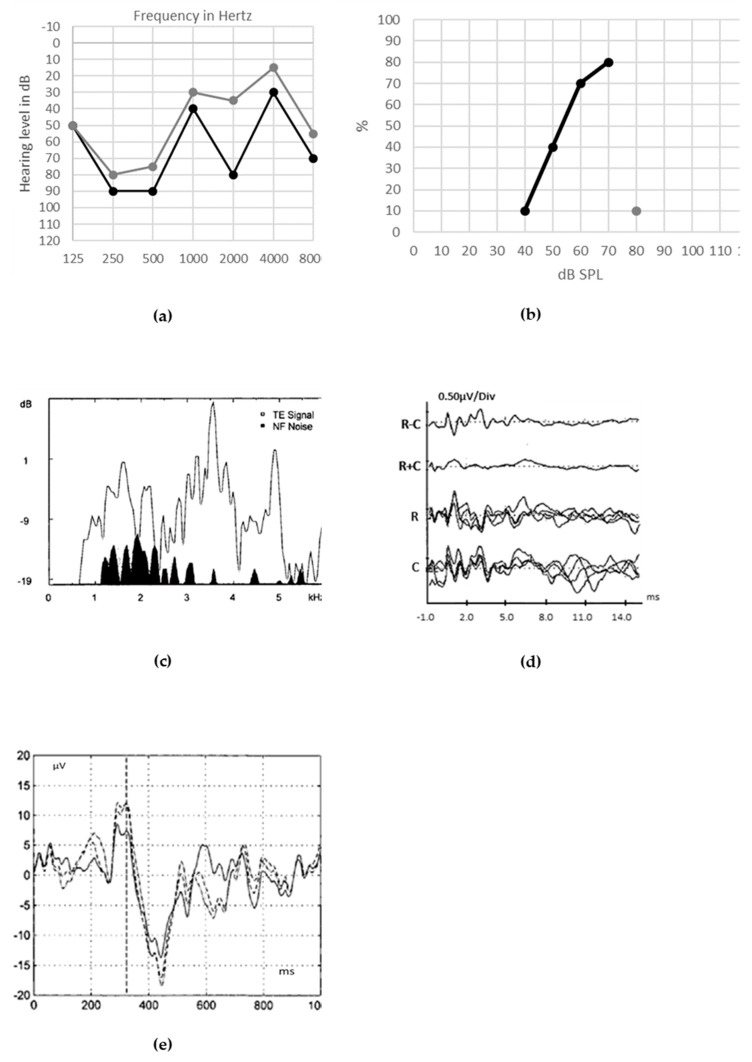
Audiological assessment in a subject with Brown–Vialetto–Van Laere syndrome-related auditory neuropathy spectrum disorder (ANSD), resulting in neural impairment of the 8th and 12th cranial nerves and the optic nerves. Unaided pure-tone audiometry ((**a**), both ears) shows bilateral hearing loss mainly for low frequencies. Unaided free field speech discrimination is poor ((**b**), 10% gray dot at 80 dB). Aided tonal and speech perception outcomes with acoustical hearing aids remain poor and comparable to unaided perception. The latter clearly improves after cochlear implantation (CI) ((**b**)**,** black line). Otoacoustic emissions (OAEs) are present ((**c**), spectral analysis of OAEs in the white area, compared to noise in the black area), whereas auditory brainstem responses (ABRs) synchronization was abnormal with no wave V but a clear cochlear microphonic (CM) in both rarefaction (R) and condensation (C) polarities and in the subtraction R − C (**d**). Cortical auditory evoked responses (**e**) show a P3 complex after stimulation with an oddball paradigm and in selective attentive conditions only, even if no recordable N100 or P200 waves are present in responses to the frequent stimuli (not shown).

**Figure 8 jcm-09-01074-f008:**
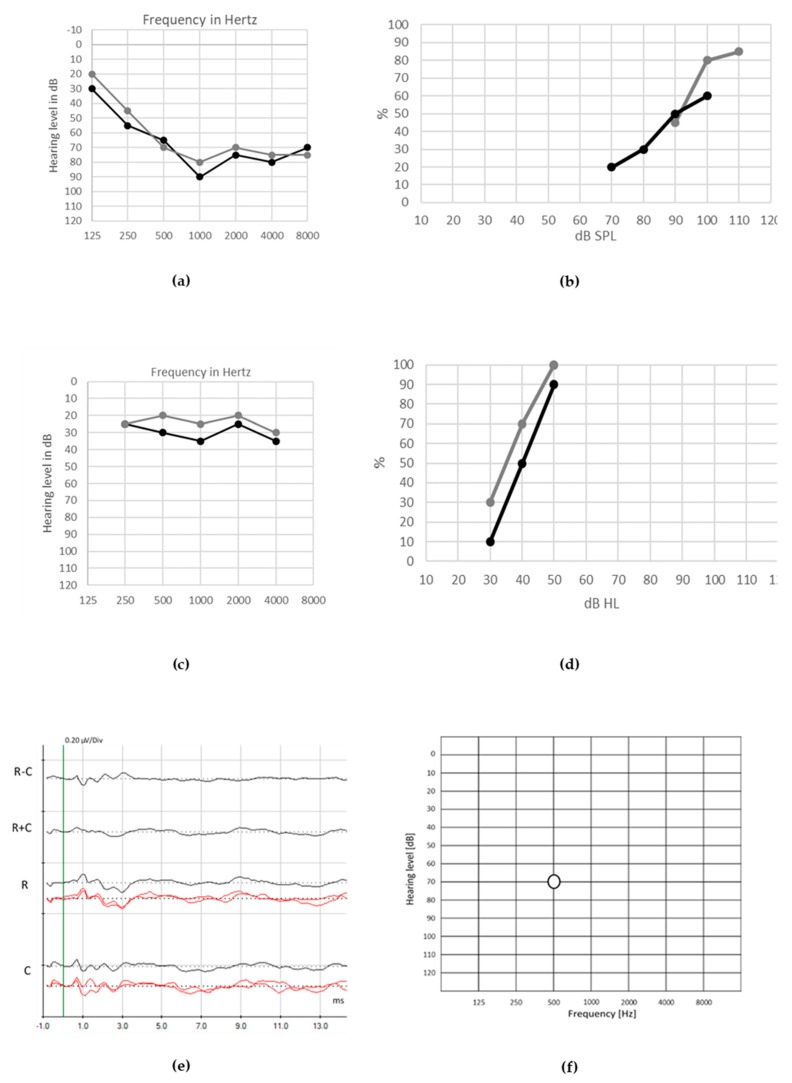
Audiological assessment in a 14-year-old child with auditory neuropathy spectrum disorder (ANSD) features occurring with severe hearing loss. Her medical history includes a 35-week preterm birth after an intrahepatic cholestasis of pregnancy, with normal birth weight but neonatal hypoxia requiring 3 weeks of stay in neonatal intensive care unit. Hearing aids were fitted in early infancy. Besides poor sounds recognition and speech perception, speech development was good in the lexical and morphosyntactic fields. Unaided tonal thresholds ((**a**), gray line for right ear, black line for left ear) and unaided speech discrimination (**b**) are poor. Aided pure-tone audiometry (**c**) and speech perception (**d**) with hearing aids are clearly improved, allowing a good development of language skills and learning abilities. Auditory brainstem responses (ABRs) elicited by clicks at 90 dB show small cochlear microphonic (CM) (**e**). Auditory steady-state responses (ASSRs) (**f**) are detected for 500 Hz at the right ear.

**Figure 9 jcm-09-01074-f009:**
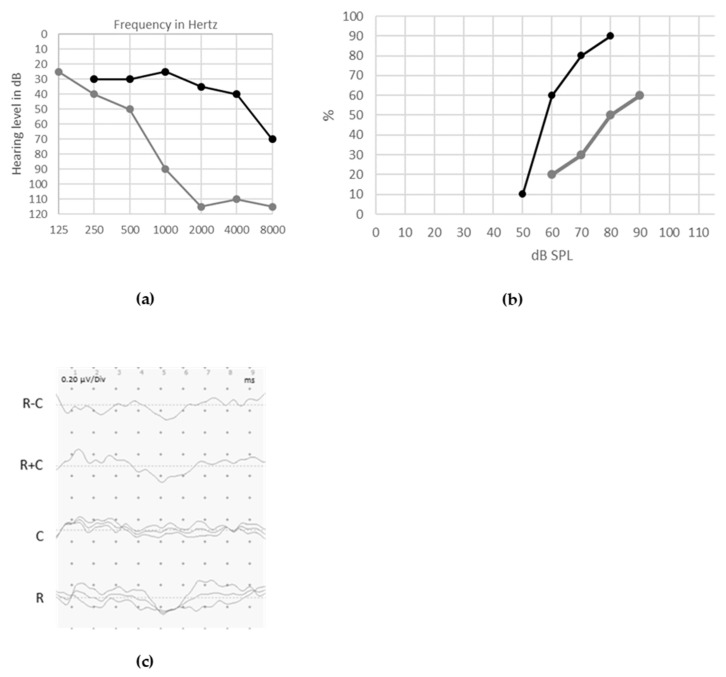
Audiological assessment in a 14-years-old male with auditory neuropathy spectrum disorder (ANSD). His medical history includes neonatal hyperbilirubinemia and low birth weight. Despite early hearing aids fitting, speech development was delayed. He shows residual pure-tone hearing thresholds ((**a**), gray line) and poor speech discrimination ((**b**), gray line) in the left ear. After left-ear cochlear implantation, aided tonal ((**a**), black line) and speech ((**b**), black line) thresholds show good auditory outcomes. Auditory brainstem responses (ABRs) at left ear are absent (**c**). However, speech perception in noise, such as during school activities, remains poor.

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
