# Peer review of "Auditory Neuropathy Spectrum Disorders: From Diagnosis to Treatment: Literature Review and Case Reports"

_jcm, 2020, doi:10.3390/jcm9041074_

Round 1

Reviewer 1 Report

The manuscript “Auditory Neuropathy Spectrum Disorders: From Diagnosis to Treatment” was intended to review the field of the auditory neuropathies. The intention is good, but the execution is not that good.

This reviewer found 30 reviews regarding this topic; 19 of them published in the last five years. It would be of benefit for the manuscript in review to specify what is the unique aim of this review and what distinguishes it from the previous reviews.

The second major problem is the type of review, which authors have not stated. It is a literature review, but it also contains the authors own case reports (or?). Would it be of advantage specifying it in the title – perhaps as a subtitle? I think – yes. Something like “Auditory Neuropathy Spectrum Disorders: From Diagnosis to Treatment. Literature review and case reports”.

As for the case reports – they should be adequately reported, including full medical history.

As for the literature review – please specify which kind of review has been performed (for the types of literature review, please see the attached publication).

Following factual errors have been found:

“IHCs is a mechanosensory structure that converts mechanical deflections…”. An inner hair cell is a cell and not a structure.

“From the IHCs row peripheral axons proceed to cell bodies of the spiral ganglion neurons in the modiolus”. Inner hair cells do not have axons. The axons belong to the spiral ganglion neurons.

“Most of sensory impairments of the organ of Corti, such as tectorial membrane, OHCs, IHCs, supporting cells or stria vascularis impairments….” – What did you mean when writing that?

Other issues:

1. Please introduce a new paragraph with abbreviations used in the review.

2. The manuscript contains several hard-to-read sentences. Please revise. Examples (there are several sentences that are har-to-read):

“The diagnosis is based on the preservation of outer hair cells (OHCs) responses, like otoacoustic emissions (OAEs) and/or cochlear microphonic (CM), associated with altered Auditory Brainstem Responses (ABRs) such as reduced or absent wave V [1–4].”

“The clinical profiles of ANSD are largely heterogeneous because they may result from both syndromic and non–syndromic genetic and/or environmental aetiologies, structural anomalies such as cochlear nerve hypoplasia or aplasia, variable lesion sites and congenital-neonatal or lately acquired dysfunctions.”

3. The manuscript contains spelling and syntax mistakes that should be corrected:

(…) as for the frequent limited benefits of acoustic hearing aids.

several Authors

difficult (?) auditory conditions

new-borns

OTOF were found

recurrent acute impairment from infancy have also been described

sensori-motor

Histological analysis of temporal bone have shown

Peroxisophagy

In Pejvakin-deficient

they need deeper changes to detect

“speech perceptual” should be “perceptual speech”

4. The authors use inconsistent hyphenation:

pre-term or preterm?

pre-synaptic or presynaptic?

5. Archaic expression:

“aforementioned components”

Factual errors:

“IHCs is a mechanosensory structure that converts mechanical deflections…”. An inner hair cell is a cell and not a structure.

“From the IHCs row peripheral axons proceed to cell bodies of the spiral ganglion neurons in the modiolus”. Inner hair cells do not have axons. The axons belong to the spiral ganglion neurons.

“Most of sensory impairments of the organ of Corti, such as tectorial membrane, OHCs, IHCs, supporting cells or stria vascularis impairments….” – What did you mean when writing that?

Author Response

The manuscript “Auditory Neuropathy Spectrum Disorders: From Diagnosis to Treatment” was intended to review the field of the auditory neuropathies. The intention is good, but the execution is not that good.

This reviewer found 30 reviews regarding this topic; 19 of them published in the last five years. It would be of benefit for the manuscript in review to specify what is the unique aim of this review and what distinguishes it from the previous reviews.

The second major problem is the type of review, which authors have not stated. It is a literature review, but it also contains the authors own case reports (or?). Would it be of advantage specifying it in the title – perhaps as a subtitle? I think – yes. Something like “Auditory Neuropathy Spectrum Disorders: From Diagnosis to Treatment. Literature review and case reports”.

We thank the reviewer for the remarks.

We have added in the text that “ This article provides a literature review on ANSD management with a special focus on its aetiologies, audiological assessment, and rehabilitation. In order to describe the variability of physiopathological and clinical features of ANSD, we used a conceptual approach based on the site of the lesion, from the more distal to the proximal site of the sensorineural pathway. Moreover, few own case reports have been described.” (page 1 line 95-98)

As for the case reports – they should be adequately reported, including full medical history.

We now provide medical history in the text as requested.

As for the literature review – please specify which kind of review has been performed (for the types of literature review, please see the attached publication).

We now specify in the title the type of the article, as suggested: “Auditory Neuropathy Spectrum Disorders: from diagnosis to treatment. Literature review and case reports.”

Following factual errors have been found:

“IHCs is a mechanosensory structure that converts mechanical deflections…”. An inner hair cell is a cell and not a structure.

We replaced it with the following : “The IHCs are  mechanosensory cells that convert mechanical deflections of the hair bundle at their apex into a molecular signal suitable for triggering the electrical firing of neuronal fibres.” (page 3 lines 123 to 125).

“From the IHCs row peripheral axons proceed to cell bodies of the spiral ganglion neurons in the modiolus”. Inner hair cells do not have axons. The axons belong to the spiral ganglion neurons.

We replaced it with the following : “From the synapse with IHCs, peripheral axons of SNGs proceed in the modiolus of the cochlea and continue as proximal axons towards the midbrain.” (page 4 line 144).

“Most of sensory impairments of the organ of Corti, such as tectorial membrane, OHCs, IHCs, supporting cells or stria vascularis impairments….” – What did you mean when writing that?

We replaced it with the following : “Most deficits caused by lesions of other cochlear components, such as tectorial membrane, OHCs, IHCs, supporting cells or stria vascularis impairments, …”

Other issues:

  1. Please introduce a new paragraph with abbreviations used in the review.
    We introduced it before the References (pag 21).
  2. The manuscript contains several hard-to-read sentences. Please revise. Examples (there are several sentences that are har-to-read):

“The diagnosis is based on the preservation of outer hair cells (OHCs) responses, like otoacoustic emissions (OAEs) and/or cochlear microphonic (CM), associated with altered Auditory Brainstem Responses (ABRs) such as reduced or absent wave V [1–4].”

We replaced it with the following : “Its diagnosis is in part based on evidence of altered neural processing such abnormal Auditory Brainstem Reponses (ABRs), with a reduced or absent wave V, despite evidence of preserved outer hair cells (OHCs) responses, such as Otoacoustic Emissions (OAEs) and/or cochlear microphonic (CM).” (page 2, line 50)

 “The clinical profiles of ANSD are largely heterogeneous because they may result from both syndromic and non–syndromic genetic and/or environmental aetiologies, structural anomalies such as cochlear nerve hypoplasia or aplasia, variable lesion sites and congenital-neonatal or lately acquired dysfunctions.”

 We replaced it with the following : “The clinical profiles of ANSD are largely heterogeneous, according to the variety of possible aetiologies. ANSD may results from both syndromic and non–syndromic genetic anomalies, environmental causes, as well as structural anomalies. Among these, cochlear nerve hypoplasia or aplasia, variable lesion sites and congenital-neonatal or lately acquired dysfunctions have been described.” (page 2 line 78)

  1. The manuscript contains spelling and syntax mistakes that should be corrected:

(…) as for the frequent limited benefits of acoustic hearing aids.
Replaced with “frequently limited benefits of acoustic hearing aids”

several Authors.
Capital letter corrected.

difficult (?) auditory conditions.
Replaced with “Challenging”.

new-borns.
Corrected as suggested.

OTOF were found.
Corrected as suggested.

recurrent acute impairment from infancy have also been described.
Replaced with the following : “recurrent acute exacerbation of the impairment from infancy have also been described” (Page 7, line 287).

sensori-motor.
Corrected as suggested

Histological analysis of temporal bone have shown.
Spelling corrected in “has”

Peroxisophagy.
Replaced with “pexophagy”.

In Pejvakin-deficient :
We replaced with the following : “the viral transduction in mice with a deficiency of pejvakin has been demonstrated to completely restore the disrupted peroxisomes proliferation triggered by pejvakin mutations, preventing the damage from oxidative stress” (Page 10, line 366).

they need deeper changes to detect.
“Deeper” has been replaced with “larger”.

“speech perceptual” should be “perceptual speech”.
Corrected as suggested.

  1. The authors use inconsistent hyphenation:

pre-term or preterm?

pre-synaptic or presynaptic?

Corrected as suggested.

  1. Archaic expression:

“aforementioned components”

Replaced with above-mentionned.

Reviewer 2 Report

In this review, De Siati et al. provide a broad overview of auditory neuropathy spectrum disorder. Details include historical background, audiologic evaluation, genetics, and treatment. 

Overall this is an excellent review. There is a lot of ongoing work in the field and so it is very timely. The references are excellent. The figures are helpful in understanding the text.

I only have a few minor points:

1) The manuscript could use minor English-language editing for grammatical errors (i.e. page 2 line 79 "IHCs is a mechanosensory structure ..."

2) All human gene names should be capitalized and italicized (i.e. OTOF)

3) Figure 3: What are the number of patients included in this figure and what are their ages? More details are required in the legend to better understand the figure.

4) Subheadings should be italicized or numbered for clarity (i.e. page 4 line 161)

Author Response

We thank the reviewer for the remarks.
Please find here, below the original review report, the correction as suggested.

In this review, De Siati et al. provide a broad overview of auditory neuropathy spectrum disorder. Details include historical background, audiologic evaluation, genetics, and treatment.

Overall this is an excellent review. There is a lot of ongoing work in the field and so it is very timely. The references are excellent. The figures are helpful in understanding the text.

I only have a few minor points:

1) The manuscript could use minor English-language editing for grammatical errors (i.e. page 2 line 79 "IHCs is a mechanosensory structure ..."

- Corrected as suggested.

2) All human gene names should be capitalized and italicized (i.e. OTOF).

- Corrected as suggested.

3) Figure 3: What are the number of patients included in this figure and what are their ages? More details are required in the legend to better understand the figure.

- We add medical history in the text.

4) Subheadings should be italicized or numbered for clarity (i.e. page 4 line 161)

  • Corrected as suggested.

This manuscript is a resubmission of an earlier submission. The following is a list of the peer review reports and author responses from that submission.